



# Tidal mixing of estuarine and coastal waters in the Western English Channel controls spatial and temporal variability in seawater $CO_2$

Richard P. Sims[1], Michael Bedington[2], Ute Schuster[3], Andrew J. Watson[3], Vassilis Kitidis[2], Ricardo Torres[2], Helen S. Findlay[2], James R. Fishwick[2], Ian Brown[2], Thomas G. Bell[2]

[1]Department of Geography, University of Calgary, Calgary, T2N 1N4, Canada
[2]Plymouth Marine Laboratory, Plymouth, PL1 3DH, United Kingdom
[3]Department of Geography, University of Exeter, Exeter, EX4 4QE, United Kingdom

*Correspondence to*: Thomas G. Bell (tbe@pml.ac.uk)

**Abstract.** Surface ocean $CO_2$ measurements are used to compute the oceanic air–sea $CO_2$ flux. The $CO_2$ flux component from rivers and estuaries is uncertain. Estuarine and coastal water carbon dioxide ($CO_2$) observations are relatively few compared to observations in the open ocean. The contribution of these regions to the global air–sea $CO_2$ flux remains uncertain due to systematic under-sampling. Existing high-quality $CO_2$ instrumentation predominantly utilise showerhead and percolating style equilibrators optimised for open ocean observations. The intervals between measurements made with such instrumentation make it difficult to resolve the fine-scale spatial variability of surface water $CO_2$ at timescales relevant to the high frequency variability in estuarine and coastal environments. Here we present a novel dataset with unprecedented frequency and spatial resolution transects made at the Western Channel Observatory in the south west of the UK from June to September 2016, using a fast response seawater $CO_2$ system. Novel observations were made along the estuarine–coastal continuum at different stages of the tide and reveal distinct spatial patterns in the surface water $CO_2$ fugacity ($fCO_2$) at different stages of the tidal cycle. Changes in salinity and $fCO_2$ were closely correlated at all stages of the tidal cycle and suggest that the mixing of oceanic and riverine end members determines the variations in $fCO_2$. The observations demonstrate the complex dynamics determining spatial and temporal patterns of salinity and $fCO_2$ in the region. Spatial variations in observed surface salinity were used to validate the output of a regional high resolution hydrodynamic model. The model enables a novel estimate of the air–sea $CO_2$ flux in the estuarine–coastal zone. Air–sea $CO_2$ flux variability in the estuarine–coastal boundary region is dominated by the state of the tide because of strong $CO_2$ outgassing from the river plume. The observations and model output demonstrate that undersampling the complex tidal and mixing processes characteristic of estuarine and coastal environment bias quantification of air-sea $CO_2$ fluxes in coastal waters. The results provide a mechanism to support critical national and regional policy implementation by reducing uncertainty in carbon budgets.



# 1 Introduction

The ocean has taken up about a quarter of anthropogenic carbon dioxide ($CO_2$) emissions to date, absorbing approximately 2.5 Pg C $yr^{-1}$ (Friedlingstein et al., 2019). Global observations of the partial pressure of $CO_2$ in seawater ($pCO_2$) are stored in the SOCAT database (https://www.socat.info/)(Bakker et al., 2016). The latest version of SOCAT (v2021), contains ~32.7

million $pCO_2$ measurements, with the majority (~>75%) of data collected in the open ocean (Laruelle et al., 2018). The carbon cycle in continental shelf waters has been extensively studied in recent years (Laruelle et al., 2018;Robbins et al., 2018;Kahl et al., 2017;Ahmed and Else, 2019;Laruelle et al., 2017b;Kitidis et al., 2019;Fennel et al., 2019). Whilst the surface area of coastal, estuarine and continental shelf waters make up 7.5% of global waters (~2.7 x $10^7$ $km^2$) (Cai, 2011), continental shelves alone have been shown to take up a substantial proportion 0.20–0.25 Pg C $yr^{-1}$ (8-10%) of global ocean

uptake (Laruelle et al., 2018;Chen et al., 2013;Cai, 2011;Bauer et al., 2013;Laruelle et al., 2014). Estuaries are net heterotrophic and thus typically oversaturated with $pCO_2$ (Laruelle et al., 2017a). The global air–water $CO_2$ flux from estuaries is thus into the atmosphere and this flux is currently perceived to roughly offset the $CO_2$ uptake by the continental shelves (Cai, 2011;Borges et al., 2006). However, the global estuarine air–water $CO_2$ flux remains highly uncertain, with estimates ranging from 0.09 to 0.78 Pg C $yr^{-1}$ (Cai, 2011;Laruelle et al., 2010;Chen and Borges, 2009;Chen et al.,

2013;Resplandy et al., 2018).

Observations of $pCO_2$ in estuarine waters included in SOCAT have only been made from a limited number of research vessels. Most research ships have drafts that limit their ability to navigate safely in waters with shallow, irregular topography. Transects of many large rivers by shallow-bottom boats identify a $CO_2$ concentration gradient between the mouth and source, with high $pCO_2$ (~10,000 ppm) upriver and $pCO_2$ levels similar to seawater close to the river mouth

(Borges et al., 2018;Joesoef et al., 2015;Macklin et al., 2014;Bozec et al., 2012;Volta et al., 2016;Cai, 2011;Jeffrey et al., 2018b). Assessment of $pCO_2$ in inland waters, determined from measurements of total alkalinity (TA) and dissolved inorganic carbon (DIC), is also highly uncertain, due to limited carbonate buffering capacity at low pH, and because acids in terrestrial organic matter make an unknown contribution to the TA anions (Abril et al., 2015).

The zone where estuarine waters meet coastal waters is a physically dynamic system influenced by riverine outflow, winds,

waves, and tidal cycles, hence $pCO_2$ is likely to vary where estuarine water and continental shelf water interact, yet there has been relatively little attention given to air–water $CO_2$ fluxes in this zone (Cai, 2011). One holistic modelling study has tried to understand the coastal system in Eastern North America, encompassing tidal wetlands, estuaries and continental shelves (Najjar et al., 2018). The study demonstrated that, despite estuaries representing <10 % of the domain area, interactions with shelf seas must be considered as the estuarine waters gave off more $CO_2$ than the shelves drew down. Eulerian studies in the

estuarine zone have identified large tidal signals in $pCO_2$ with data collected from research vessels (Borges and Frankignoulle, 1999;Jeffrey et al., 2018a) and from moorings or fixed sites (Dai et al., 2009;Jeffrey et al., 2018a;Li et al., 2018;Bakker et al., 1996;Call et al., 2015;Ferrón et al., 2007;Santos et al., 2012). $pCO_2$ levels at the mouth of different rivers have been observed to co-vary with changes in river flow rate and with the tidal cycle (Ribas-Ribas et al., 2013;Canning et



al., 2021;Najjar et al., 2018;Frankignoulle et al., 1996).  Transects underway $CO_2$ measurements have not been regularly
repeated through the transitional zone where estuaries connect with continental shelf waters.

Here we present weekly surface water $fCO_2$ observations along a transect from estuarine waters into continental shelf waters
in the English Channel between April and September 2016 (Figure 1). The $CO_2$ measurement system was capable of high
spatial (~0.2 km) and temporal resolution (~ 48 s) revealing substantial variations in surface $fCO_2$.  Our aim was to examine
in detail the transition zone between rivers, estuaries and the coastal zone in order to investigate variations in surface $CO_2$
and potential drivers of this variation.

We then used the observed relationship in the $CO_2$ and salinity data together with the output of a high resolution
hydrodynamic model of the estuary and nearshore region to estimate the spatial heterogeneity in $fCO_2$ and air/water $CO_2$
flux.

## 2. Study location and physical setting

This study was conducted in the English Channel off the coast of Plymouth (south-west UK) at the Western Channel
Observatory (WCO, see Figure 1a), which  includes the oceanographic station L4 (50.251°N, 4.221°W; (Smyth et al., 2010a)
and the Penlee Point Atmospheric Observatory (PPAO, 50.319°N, 4.193°W;(Yang et al., 2019;Yang et al., 2016) among
other routine monitoring activities and assets. The PPAO is a coastal land-based observatory on the Rame Head peninsula at
the entrance to the Plymouth Sound. Station L4 is a coastal site ~8 km from PPAO and is a focal point of the ongoing WCO
time series.

L4 is seasonally stratified between late April and early October (Smyth et al., 2010b). The onset of stratification typically
drives a diatom dominated spring bloom in early April. Nitrogen limitation later in the year favours a summertime
dominance of smaller plankton (Widdicombe et al., 2010).   High river discharge rates in July 2006 were caused by several
days of high rainfall and resulted in ~0.5 psu reduction in surface salinity at L4 (Rees et al., 2009).  A prominent feature of
the coastal region around WCO is the coastal/tidal current that entrain buoyant freshwater from the River Tamar outflow
with prominent frontal features (Uncles and Torres, 2013). The River Tamar is a large source of freshwater to the region
despite being a relatively small river (Uncles et al., 2015). The coastal current moves along the  west coast of Plymouth
Sound adjacent to the Plymouth Breakwater and toward PPAO before following the coastline towards Rame Head peninsula
(Uncles et al., 2015;Siddorn et al., 2003).

The annual cycle of L4 surface water $pCO_2$ is mainly determined by biological activity, with the spring bloom depleting
$pCO_2$ in April and May (Torres et al., 2020). Seawater $pCO_2$ increases to pre-bloom levels throughout the summer and is at
equilibrium or slightly oversaturated in the autumn and winter (Kitidis et al., 2012). Surface water $pCO_2$ decreases with
distance offshore at all times of the year, with the greatest variability within 10 km of the coast (Kitidis et al., 2012).
Changes in surface $pCO_2$ >40 ppm have been observed during a 24 hour eulerian study at L4 (Litt et al., 2010). Direct



methane flux measurements made at the PPAO indicate that the waters around the Rame Head peninsula are influenced by
the interaction between tidal cycles and the Tamar freshwater outflow (Yang et al., 2016).

## 3. Methods

**Sampling approach:**

Transects between the Breakwater in the Plymouth Sound and Station L4 were conducted weekly where possible during the
study period (Figure 1b). Transects took place between 09:00 and 15:00 hrs local time using the RV *Plymouth Quest*.  15
transects were conducted on 12 non-sequential days between 10$^{th}$ June and 21$^{st}$ September 2016. RV *Plymouth Quest* takes
~40 minutes to travel directly to L4 during transects, except during a number of voyages when the ship was redirected into
Cawsand Bay and/or closer to the PPAO. The underway seawater system on the ship has an intake at 3 m depth, supplying
seawater for measurements of $pCO_2$, sea surface temperature (SST) and surface salinity (SBE45; Seabird Scientific, USA).
The underway system is turned off when the ship is shore side of the Breakwater to reduce the risk of heavy biofouling and
the intake of large quantities of sediment and coastal debris.

A rigid inflatable boat (*PML Explorer*) was used to sample the River Tamar on the 1$^{st}$ October 2014. SST and salinity were
measured with a portable CTD package (SeaCat CTD 19+; SeaBird Scientific) and seawater was sampled at 9 stations along
a ~25 km transect between the Breakwater and Calstock Slip (Figure 1b, 50°29.732'N, 4°12.408'W). Seawater was collected
for laboratory analysis of total alkalinity (TA) and dissolved inorganic carbon (DIC) following the best practice (SOP 1 of
Dickson et al. (2007)). The salinity (35.16 PSU) and $pCO_2$ (410.93ppm) were measured by the RV *Plymouth Quest* on the
29$^{th}$ September 2014 at station L4.

**Seawater CO2 and carbonate system analyses:**

Two independent $CO_2$ systems with different equilibrator designs were used to measure seawater $CO_2$ on the RV *Plymouth*
*Quest*. The PML-Dartcom Live $pCO_2$ system (Dartcom systems Inc, UK) is installed permanently  installed to measure
ocean surface $CO_2$ at L4, and utilises a vented showerhead equilibrator with an equilibration time of 8 minutes (4 e-folding
times). This was setup with a sampling frequency of 27 minutes (Kitidis et al., 2012). The showerhead system also measured
atmospheric  $pCO_2$ every  27 minutes. Additionally, a high-frequency and high-resolution membrane $CO_2$ system (Sims et al.
(2017) was installed during the study period.  This system utilises a membrane equilibrator with a fast response time of 48
seconds (2 e-folding times) and has a high sampling frequency (1 Hz). Both $CO_2$ systems were calibrated with the same
secondary standards (263.04 and 483.36ppm), traceable to WMO standards by cross-calibration at PML against  National
Oceanic and Atmospheric Administration Global Monitoring Laboratory (NOAA GML, USA) certified standards (244.91,
388.62 and 444.40 ppm). The membrane system was calibrated pre- and post-voyage (~4–5 hours apart) and the showerhead
system calibrated hourly while the underway system was running.





The membrane and Dartcom systems used non-dispersive infrared gas analysers (Model 840B and 7000, respectively; Licor, Inc., USA) to measure gas phase $CO_2$ mixing ratio ($xCO_2$) in the equilibrated gas exiting the membrane or shower-head equilibrators. The fugacity of $CO_2$ inside the equilibrator ($fCO_{2(eq)}$) and the fugacity of $CO_2$ in seawater ($fCO_{2(sw)}$) were calculated using  measured SST, surface salinity, equilibrator pressure and equilibrator temperature, using  equations listed in SOP 5 (Dickson et al., 2007). Lag-time correlation analysis between SST and the equilibrator temperature identified that a

-79 sec adjustment should be made to the $fCO_2$ measurement time. The $fCO_2$ time adjustment accounted for the time taken for water to enter the seawater intake beneath the hull of the ship and arrive at the showerhead and membrane equilibrators. The flux (F) of $CO_2$ ($mmol\ m^{-2}\ d^{-1}$) is computed  following (Wanninkhof, 2014a):

$$F_{(sea-air)} = k_W\ k_0\ \Delta fCO_2\ SF \tag{1}$$

$k_w$ is the water phase gas transfer velocity ($cm\ hr^{-1}$; Equ. 2), $k_0$ is the solubility of $CO_2$ in seawater ($mol\ L^{-1}\ atm^{-1}$) from

Weiss (1974),  $\Delta fCO_2$ is the difference in fugacity between the atmosphere and ocean ($\mu atm$; Equ. 3), and SF is a scaling factor of 0.24 to express the result flux F in units of ($mmol\ m^{-2}\ d^{-1}$).

The gas transfer velocity is computed following Nightingale et al. (2000):

$$k_W = (0.222\ (U_{10})^2 + 0.333\ (U_{10}))\ (Sc/660)^{-1/2} \tag{2}$$

where $U_{10n}$ is the mean wind speed at 10m height and for neutral conditions, Sc is the scaling by unitless Schmidt number of

660 (Wanninkhof, 2014b). $U_{10n}$ measurements were taken from the L4 buoy (Smyth et al., 2010a).

$\Delta fCO_2$ is here defined as the difference between the seawater interface fugacity and the fugacity of air:

$$\Delta fCO_2 = fCO2_{(sw)} - fCO_{2(air)} \tag{3}$$

To account for the cool and salty layer at the atmosphere ocean interface, -0.17°C and +0.1 PSU were added to the in situ temperature and salinity when calculating seawater $fCO_2$ at the interface and for the estimation of air–sea $CO_2$ flux (Woolf et

al., 2019).

TA samples were measured by open cell potentiometric titration with 0.1M hydrochloric acid on a Total Alkalinity Titrator (Model AS–ALK2; Apollo SciTech, Inc. USA) following SOP 3b in Dickson et al. (2007). DIC was measured using an Apollo SciTech DIC analyser (Model  AS-C3; Apollo SciTech, Inc., USA) following SOP 2 in Dickson, Sabine et al. (2007), by acidifying samples with excess 10% phosphoric acid and using nitrogen gas to transfer liberated $CO_2$ to an

infrared gas analyser (Model Licor 7000, Licor Inc., USA). These two measurement systems are described in more detail in (Kitidis et al., 2017). $fCO_2$ was calculated from TA and DIC using CO2SYS in Matlab (Van Heuven et al., 2011) with the carbonic acid dissociation constants of Mehrbach (1973) refit by Dickson and Millero (1987) and the hydrogen sulphate dissociation constant of Dickson (1990).





**Numerical model:**

The hydrodynamics of the WCO were modelled following Uncles et al. (2020) using an implementation of the Finite-Volume Community Ocean Model (FVCOM;(Chen et al., 2003)). The model domain (~ 49.7° to 50.6° N and ~ 4.8° to 3.8° W) encompasses station L4, PPAO, Plymouth Sound, and the estuary of the River Tamar including its major tributaries. FVCOM utilises an unstructured grid consisting of a mesh of variable resolution triangles, which allows the representation of complex coastlines and higher resolution in areas of interest whilst remaining computationally feasible. The resolution of the

model is ~600 m at L4, becoming finer towards the PPAO and Plymouth Sound (~85 m), with highest resolution around the upper River Tamar channel (~40 m). The vertical system is terrain following sigma coordinates with 24 equally spaced layers. Horizontal mixing is parameterised through the (Smagorinsky, 1963) scheme and vertical turbulence closure through an updated version of the MY 2.5 scheme(Mellor and Yamada, 1982), with mixing coefficients of 0.2 and $1\times10^{-5}$ respectively.

Water depth within the model domain uses the EMODNET bathymetry product with a nominal resolution of 1/16 degree. In the estuary and nearshore areas (< 20 m depth) the data have been complemented with local data sources of Lidar, single and multi-beam surveys accessed through the Coastal Channel Observatory (CCO, https://www.channelcoast.org/). The CCO data was re-projected from its original projection (OSGB) to WGS84 and concatenated and averaged into a 20 m regular grid using the Generic Mapping Tools (GMT 5.3.2) software (http://gmt.soest.hawaii.edu/). The merged dataset was processed

using the ROMS toolbox for bathymetry processing downloaded from https://github.com/dcherian/tools. The scattered bathymetry was interpolated to 25 m and smoothed iteratively to achieve a Haney number less than 2 (Haney, 1991).

Lateral boundary conditions are provided as a one-way direct nesting at hourly resolution from a larger FVCOM model covering the west UK shelf (Cazenave et al., 2016). The surface atmospheric forcing is from the NCEP GFS 6 hourly historical product, which is downscaled to ~3 km resolution using a triple nested implementation of the Weather Research

and Forecasting (WRF) model (Skamarock et al., 2008). The freshwater input flux for 11 rivers in the domain was obtained from daily river gauge data from the National River Flow archive (https://nrfa.ceh.ac.uk). The temperature of freshwater inputs was determined using a regression model on the WRF 2m air temperature, which was trained on temperature records from the UK Environment Agency's Freshwater River Temperature Archive (Orr et al., 2010) and data from the Westcountry river trust.

The model was run for the period between March 2016 and November 2016, with instantaneous values output hourly. Only the uppermost layer of the FVCOM output is used in the analysis below. The thickness of this layer varies between 0.05 m and 5m  across the domain (because the model uses terrain-following coordinates), and the significance of this and how it relates to the 3 m underway  measurement and air–sea $CO_2$ fluxes is discussed below.





## 4. Results

**Showerhead vs membrane comparison:**

Surface $fCO_2$ was measured for several hours while RV *Plymouth Quest* was stationary at L4 in between outbound and return transects. The showerhead system sampled continuously from the fixed seawater intake (3 m depth). The membrane equilibrator system was switched to sample via tubing from a near surface ocean profiler (Sims et al., 2017) when at L4 (Figure S1). The two $CO_2$ systems showed good agreement when the surface ocean profiler was between 2.5m and 3.5m, despite sampling slightly different water through different tubing (Figure 2). The mean residual was 2.77 µatm and the RMSE was 6.9 µatm when sampling on station (Figure 2), this is similar to the ±2 µatm difference that has been observed during other seawater $CO_2$ intercomparison exercises (Körtzinger et al., 2000;Ribas-Ribas et al., 2014). The $CO_2$ systems both sampled from the underway seawater supply of the ship during the L4 to Breakwater transects, and $fCO_2$ was estimated from $pCO_2$ using the underway SST and salinity. The 11 coincident $fCO_2$ measurements during the transects had a mean residual of 2.88 µatm and a RMSE of 27.1 µatm (Figure 2). The largest differences between the $CO_2$ systems tend to occur when the ship was not on station at L4.

**Spatial and temporal variation in salinity, SST and $fCO_2$:**

Sea surface salinity at L4 during the June to October study period does show small variation (mean ±SD = 35.15±0.08 psu; Figure 3a). The L4 salinity range during the study period is within the range of previous years (Smyth et al., 2010b). L4 salinity decreased intermittently, with a maximum reduction of 0.68 psu on 9[th] August. The salinity measurements at L4 contrast with the large variability in Breakwater salinity during the study period (34.17–35.23 psu). The average salinity difference between the Breakwater and L4 is 0.36 psu. The salinity ranges observed at L4 and at the Breakwater are similar to model results during a tidal cycle (Uncles et al., 2015).

Coastal SSTs (Figure 3b) followed the seasonal warming pattern slowly increasing from 13.9°C on 10[th] June to 16.7°C on 21[st] September. The warming trend was in broad agreement with SST trends observed during previous years (Smyth et al., 2010b). The SST variation along each transect was typically small (<1°C) compared to the SST change over the study period (2.8°C). The smallest temperature difference between L4 and the Breakwater (0.025°C) occurred on 15[th] September. The largest temperature differences occurred during early summer (1.20°C, 0.63°C, 0.59°C and 0.65°C on 10[th], 15[th], 22[nd] and 30[th] June, respectively).

L4 surface $fCO_2$ increased from 355 µatm to 420 µatm between 10[th] June and 21[st] September (Figure 3c). The average $fCO_2$ difference between L4 and the Breakwater was 20 µatm. Abrupt changes in $fCO_2$ were observed along the transects, generally close to the Breakwater (>4 km from the L4 station). $fCO_2$ in waters within 4 km of the Breakwater ranged between 338 and 440 µatm during the study period. $fCO_2$ measurements within 2 km of the PPAO also varied considerably during some of the transects; for example, $fCO_2$ varied between 337 and 434 µatm during the transect past the PPAO on 7[th]





July. The variability in membrane equilibrator fCO$_2$ measurements agreed with previous observations in the region (Kitidis et al., 2012).

When transects were ordered by stage along the 12 hour tidal cycle, the spatial variability in salinity transects (Figure S2) appeared to correspond to the different stages of the tidal cycle. Temperature and fCO$_2$ transects can be found in the

supplementary materials (Figure S3-4). We divided our transect data according to the stage in the tidal cycle (determined from the Devonport tidal gauge station; 50.368°N, -4.185°W, Figure 1b). Transects were characterised by the time elapsed since the last period of low water (i.e. LW + X hrs) using the time at the midpoint of each transect.

Four categories were used, corresponding to 3 hr tidal periods: Low water (LW+0 hrs); Flooding tide (LW+3 hrs); High water (LW+6 hrs); and Ebbing tide (LW+9 hrs). Four transects are used below to exemplify the relationship between the

different stages of the tidal cycle and the spatial variation in fCO$_2$ (Figures 4–7).

**Example transects:**

The transect on 7[th] July (low water, LW+0 hrs) began at L4 and headed inland (Figure 4). Salinity was 35.1 psu close to L4 and declined along the transect. The low salinity was directly North of PPAO (34.7 psu), and in the shallow waters of Cawsand Bay (North of PPAO, 35.5 psu). SST was warmest near L4 and in the relatively shallow Cawsand Bay. The river

water was cooler than the L4 seawater and there was a negative gradient (0.55°C) into Cawsand Bay. fCO$_2$ was lowest (~360 μatm) when near to L4 and highest (~430 μatm) immediately South of PPAO. fCO$_2$ to the North of PPAO in Cawsand Bay was ~390 μatm, lower than waters South of PPAO.

The data collected on 15[th] June is an example from a flooding tide transect (LW+3 hrs, Figure 5). Salinity was 35.1 psu in waters close to L4 and decreased toward PPAO and the Breakwater (34.7 psu minimum). The Tamar river plume was

warmer than coastal waters in July. SST was highest (~14.1°C) close to the Breakwater and the coldest water along the transect (13.3°C) was near to L4. The difference in fCO$_2$ between L4 and the Breakwater was ~18 μatm, with higher fCO$_2$ (353 μatm) in the warm, low salinity water close to the Breakwater.

Data collected on 30[th] June is an example of a transect at high water (LW+6 hrs, Figure 6). High salinity (>35 psu) water was observed much closer to PPAO during the LW+6 hr transect than on transects during low water periods in the tidal cycle

(Figures 4&5). Salinity reduces rapidly to 34.1 psu in between the PPAO and the Breakwater. Spatial patterns similar to surface salinity are seen in SST and fCO$_2$, with the gradients the inverse of the salinity gradient. The low salinity water close to the Breakwater has higher SST and fCO$_2$ (14.8°C and 385 μatm) than the observations South of PPAO. The fCO$_2$ range on this transect was ~38 μatm, and the majority of the change was close to the PPAO. The fCO$_2$ difference between L4 and the water just South of PPAO was only ~12 μatm. These data suggest that the Tamar plume during LW+6 hrs was restricted

near to the coast and did not make a big impact upon the waters in between PPAO and L4.

Data on 10[th] June show a transect during an ebbing tide (LW+9 hrs) (Figure 7). Salinity near to L4 was 35.15 psu and declined toward the coast, interrupted by a patch of water due east of PPAO with salinity levels similar to L4. Two patches of low salinity water (34.8 psu) were encountered along the transect. One low salinity patch was just South of PPAO and the



other patch was close to the Breakwater. Spatial variation in SST and $fCO_2$ coincide with the variations in salinity, with

higher SST (~14.2°C) and $fCO_2$ (~365 µatm) in the patches of low salinity water. The $fCO_2$ difference between L4 surface

waters and the fresh, warm waters influenced by the River Tamar plume is ~25 µatm.

In summary: The variations in salinity observed in all transects (i.e. examples above and those in Supplementary Figure S2)

qualitatively agreed with output from a previous physical model of the Plymouth Sound (Siddorn et al., 2003). The

freshwater plume at low water extends out past the PPAO, whereas the plume at high water (LW+6 hrs) is restricted to close

to the breakwater (Siddorn et al., 2003). The same model on an ebbing tide predicts that two plumes of river water exit via

the channels at the east and west ends of the Breakwater.

**Relationship between salinity and $fCO_2$:**

The changes in salinity and $fCO_2$ during the example transects suggest that the changes in $fCO_2$ are driven by conservative

mixing of salt and fresh water end members. Robust $fCO_2$ and salinity relationships have been observed in large rivers like

the Amazon and Mississippi (Lefèvre et al., 2010;Cai et al., 2013). The relationship between salinity and $fCO_2$ is

complicated by the seasonal variability in $fCO_2$ and the constantly changing position of the tidally-controlled freshwater

plume. We calculated the difference between *in situ* data from transects and from measurements at L4 to account for

seasonal variations in seawater $CO_2$:

$$\xi fCO_2 = \text{Transect }_{fCO2} - \text{L4 }_{fCO2} \qquad\qquad (4)$$

$$\xi S = \text{Transect }_{Salinity} - \text{L4 }_{Salinity} \qquad\qquad (5)$$

$\xi fCO_2$ and $\xi S$ displayed an apparent relationship throughout different seasons and stages of the tidal cycle (Figure 8). $\xi S$ was

a stronger predictor of $\xi fCO_2$ than location along the transect (distance from L4).

The $fCO_2$ and salinity data diverged from the general trend when the RV *Plymouth Quest* travelled into Cawsand Bay (West

of the Breakwater) on 7[th] July (Figure 8). Uncles and Torres (2013) showed that the seawater residence time increases as you

approach the Breakwater, following that methodology the FVCOM model used here also shows this is true for Cawsand

Bay. A long residence time and a shallow water column mean that the waters in Cawsand Bay are likely to be closer to

equilibrium with the atmosphere due to relatively higher turbulent mixing from bottom stress and surface waves (Upstill-

Goddard, 2006). As Cawsand Bay is a micro-environment, the bay could be a biological hotspot driving the changes in $CO_2$.

Excluding the 7[th] July transect that went into Cawsand Bay, the linear fit to the $\xi fCO_2$ and $\xi S$ data gives $\xi fCO_2 = -39.83\ \xi S +$

5.50 ($R^2$=0.2165, N=22262, p=<0.001). The fit explained 21.6% of the variability in the data.

$fCO_2$ calculated from TA and DIC bottle data from the River Tamar on 1[st] October 2014 show a qualitatively similar

relationship between salinity and $fCO_2$ (Figure S5) for stations T3–T9. The trend continues for stations furthest upstream (T1

and T2), but does not follow the same linear relationship. A linear fit between $\xi S$ and $\xi fCO_2$ for stations T3–T9 suggests a

42.24 µatm change per unit of salinity. The bottle data indicates that the $\xi fCO_2$ and $\xi S$ relationship can be extrapolated to

lower salinities and that the relationship begins to break down at $\xi S < -4$ psu. Note that the TA/DIC data and $fCO_2$ data were





collected at different times (years and stages of seasonal cycle) and with different sampling and analytical approaches. The apparent linear relationship between $\xi fCO_2$ and $\xi S$ suggested that it can be applied to the wider coastal region. The next section assesses the utility of the high resolution model (Uncles et al., 2020) to predict coastal $fCO_2$.

**Using FVCOM to estimate coastal air–sea CO$_2$ fluxes:**

Extrapolation of the relationship between $\xi S$ and $\xi fCO_2$ across the coastal domain required that FVCOM accurately replicates available surface salinity observations at different stages of the tidal cycle. FVCOM compares very well with the spatial and temporal variability in the four example transects at different stages of the tidal cycle (Figure 9), even though the hourly resolution of the model means the model output frequency can not capture subtle changes in frontal positions that are penalised in point-to-point comparisons. Model agreement with the $\xi S$ observations from all 15 transects was also good

(RMSE=1.044), which indicates that FVCOM can be used as a reliable indication of spatial and temporal salinity changes during the observation period (June–September 2016) (Figure S6).

Modelled spatial variations in coastal surface salinity are shown in Figure 10a, which focuses on the 7[th] July transect. Equivalent plots for the other three example transects on 30[th] June, 15[th] June and 10[th] June (Figures S7 to 9). The 7[th] July transect was at low water (LW +0 hrs), and when low salinity water was exiting from either side of the Breakwater and

wrapping around the Rame Head Peninsula (Figure 10a).

FVCOM output was combined with surface measurements of $fCO_2$ at L4 (Kitidis et al., 2012) to predict coastal variability in air–sea $CO_2$ fluxes (Figure 10b). The model domain for estimated $fCO_2$ is restricted i) to cover the region inshore of L4 (>50.24°N, Figure 10b), ii) to avoid regions where gas exchange is largely caused by bottom driven turbulence (bottom depth >10 m), and iii) to avoid over-extrapolating the relationship observed in Fig 8 ($\xi S > -4$ psu, equivalent to a minimum

salinity of approximately 30 psu). The model domain excludes the upper sections of the rivers, and the upper estuary and shallow coastal waters such as Cawsand Bay. The model results suggest that the majority of the coastal waters (Plymouth Sound and surroundings) contain $fCO_2$ levels that are substantially higher (>100 µatm) than L4 at this stage of the tidal cycle (low water, LW+0 hrs).

The FVCOM model enables simulation of a full tidal cycle so that the impact on air–sea $CO_2$ flux can be visualised

throughout 7[th] July (Figure 11, equivalent figures for the examples on 30[th] June, 15[th] June and 10[th] June are in Figures S10 to S12). A constant wind speed equal to the mean across the period is applied across the model domain (surface area = 132 km$^2$) for simplicity. The flux signal co-varies with tidal height (Figure 11a-b). The flux integrated across the model domain changes substantially throughout the tidal cycle (~9.3% change). If the region of the integration is restricted further to include just the region due north of PPAO (>50.32°N) then the magnitude of the flux changes substantially, it was a weak

sink (~-65 mol hr$^{-1}$) at low tide whereas at high tide there was a substantially larger sink (~-1113.23 mol hr$^{-1}$).

Three configurations were used to calculate the flux (Figure 12) to establish whether the tidal plume had a net effect on the regional air–sea fluxes calculated on seasonal timescales between March and November 2016.



In the first configuration (L4), L4 measurements were scaled by surface area. The second configuration (this study) used the L4 measurements with the $fCO_2$ and salinity relationship determined above, applied to the spatially varying salinity field

from the FVCOM model. In the third configuration (Landschützer coastal product), the closest node (50.125°N, 4.125°W) in the widely used Landschützer coastal product (Landschützer et al., 2020) is scaled by surface area. Fluxes were computed for a fixed region of 128 $km^2$ (a slight modification of the region in Figures 10b and 11a-d) encompassing the river plume and the region behind the Breakwater. Regions met the depth and salinity criteria in the region (depth >10m, $\xi S$ > -4 psu and >50.24°N) for > 80% of the period included. The average regional $fCO_2$ (i.e. including using the spatially varying salinity),

showed undersaturation in the summer and oversaturation in the spring and autumn, they were ~30 μatm greater than the observed L4 values (Figure 12a). The $pCO_2$ in the Landschützer coastal product was much lower than the observed $fCO_2$ values at L4 in 2016 (Figure 12a). Fluxes were calculated using monthly $U_{10}$ values (monthly values were required to prevent wind speed variability overshadowing changes in the flux due to $CO_2$) from the L4 buoy and atmospheric $fCO_2$ from the RV *Plymouth Quest* for flux estimates. Temperature and salinity from the L4 mooring were used to calculate the L4 and

Landschützer flux whereas the regional flux product (this study) used the hourly temperature and salinity fields from the FVCOM model (Figure 12b). The L4 fluxes followed the same trend as previous measurements at the site: $CO_2$ uptake in the region throughout the summer and a short period of outgassing in late September (Kitidis et al., 2012). The Landschützer flux product suggested that the region is a larger sink than indicated by the measurements at L4, and does not suggest that the region is ever a source of $CO_2$ to the atmosphere. The fluxes calculated with the $fCO_2$ values predicted from salinity (this

study) suggest that the near coastal region (128 $km^2$) was a weaker sink in the summer. This study also suggests weak outgassing of $CO_2$ from the near coastal region in March, April and October. In contrast, if the L4 measurements are taken as representative of the near coastal region than the region is instead calculated as being a weak sink. As salinity is the variable driving the differences between this study and the L4 measurements in spring, it can be surmised that this is due to increased freshwater from riverine input in spring.

**5. Discussion**

Here we show large variations in $fCO_2$ in the estuarine zone using a high-frequency membrane equilibrator which could not be detected using a showerhead equilibrator. Seawater $fCO_2$ is assumed to be relatively homogeneous over large spatial scales in the open ocean (Takahashi et al., 2009). Temporal variability in open ocean waters is also typically slow, with the most rapid variations occurring over hourly or slower timescales (e.g. diurnal,(Torres et al., 2021)). The measuring

frequency of a typical showerhead seawater $CO_2$ system (~8 min) is thus sufficient to determine open ocean $fCO_2$ variability. The difference between the underway $fCO_2$ measurements made by the two systems used in this study (Figure 2) is caused by the different response times of the showerhead and membrane equilibrators and their ability to resolve the considerable horizontal variability in coastal and shelf waters (Kitidis et al., 2012).



Single seawater $CO_2$ observations are representative of $CO_2$ integrated across the region that the ship transits through whilst
equilibrating. The RV *Plymouth Quest* typically travels at ~9 knots (4.6 m/s), which is equivalent to 0.22 km in 48 s (2 times
the e-folding response time of the membrane system) and 1.1 km in 4 minutes (2 times the e-folding response time of the
showerhead system). However, the showerhead system also switches between seawater $CO_2$, $CO_2$ gas standards and
atmospheric $CO_2$, hence seawater measurements are not continuous. Routines of seawater $fCO_2$ measurements are 27 min
apart, which equates to a distance of ~7.5 km when the RV *Plymouth Quest* is moving.

We found an inverses linear relationship between salinity and $fCO_2$ and that salinity measurements also agreed well with
model salinity output. The FVCOM model output confirms that the river plume extent is highly variable in time and space,
and that oceanographic features in the region such as the river plume are ~1 km or smaller. The highly dynamic nature of the
plume means that a showerhead $CO_2$ system cannot be expected to resolve the spatial dynamics. A showerhead $fCO_2$ system
design can also be biased toward the end of the equilibration period, which may generate some measurement bias if seawater
$fCO_2$ changes rapidly during the 8 min full equilibration time (e.g. (Ribas-Ribas et al., 2014)). Complete characterisation of
$fCO_2$ in coastal waters within 10 km of land and with highly variable $fCO_2$ requires a system with a fast response equilibrator
or slower steaming.

Estuarine and continental shelf water masses determine the $fCO_2$ levels observed in this near shore study region. Shelf
waters exhibit a seasonal surface $fCO_2$ trend associated with a) cooling-enhanced solubility in late winter, b) net
phytoplankton uptake in spring/summer , c) partial re-equilibration with the atmosphere and d) mixing with $CO_2$-rich bottom
waters in autumn (Kitidis et al., 2019). The River Tamar is the other mixing end member and is typically oversaturated with
$fCO_2$ (Frankignoulle et al., 1998). The results presented in this paper also suggest that riverine $fCO_2$ may have a strong
influence on the spatial and temporal patterns in coastal waters off Plymouth. It is important to note that the relationship
between salinity and $fCO_2$ determined in this paper was derived with summer data only (due to sampling constraints), and
further work is needed to confirm if the relationship stands during the remaining period of the year (i.e. October through to
May).

The confluence of estuarine and continental shelf waters near the PPAO/Breakwater and the local circulation of surface
waters are dominated by the semi diurnal tidal cycle (Uncles et al., 2015). The rising tide after low water pushes shelf water
toward the Breakwater, whereas a falling tide after high water encourages riverine-influenced water to extend south. The
tidal influence upon coastal $fCO_2$ is evident in the 15 transects during this study, with changes of 20–40 µatm relative to L4
(Figure 8). Timeseries data presented by Borges and Frankignoulle (1999) indicate a similar magnitude (20–25 µatm) tidal
signature in coastal Belgian and Dutch waters.

Accurate $fCO_2$ data are essential when comparing $\Delta fCO_2$ with direct eddy covariance $CO_2$ flux observations and trying to
estimate the gas transfer velocity (Dong et al., 2021). Well-resolved seawater $fCO_2$ observations are needed within a coastal
eddy covariance flux footprint when the flux observations are made from a fixed platform such as the PPAO (Yang et al.,
2016). The RV *Plymouth Quest* transects through the PPAO flux footprint once a week, but the exact timing of this within a
tidal cycle is determined by when the vessel is able to exit and enter the tidally restricted harbour lock. Flux footprint



sampling is thus restricted to certain stages of the tidal cycle and it is not possible to characterise the full extent of estuarine/shelf water mixing at all stages of the tide. Autonomous platforms (moored or otherwise) instrumented with high-

frequency seawater $CO_2$ sensors would   make a crucial contribution to the current observational gap (Chavez et al., 2018;Manning et al., 2020) and achieve high resolution $fCO_2$ mapping in the coastal zone.

The salinity field from the FVCOM model is used to estimate the surface $fCO_2$ features over a complete tidal cycle and shows that the flux can change by a non-trivial amount (~10%) over a short period of time (Figure 11). These data also show that using a hydrodynamic model can extend the spatial and temporal coverage of $CO_2$ uptake and outgassing estimates in

the coastal region surrounding Plymouth made with weekly L4 measurements (Figure 12). If similar relationships can be developed for other estuarine zones, then operational dynamic models may be able to add information to flux estimates for these zones, too. The spatial resolution of the model at the surface means that it is possible that near surface features such as fresh water lenses are being overlooked by this analysis. Higher resolution modelling of the surface ocean would make future coastal products more applicable to air sea flux studies.

**6. Conclusions**

Here we show for the first time dynamic and large variations in the coastal zone using a fast response membrane equilibrator system. These gradients between estuarine waters and shelf waters were not resolved by a co-located showerhead $CO_2$ system. The gradients in salinity and $fCO_2$ fields show strong tidal influence. Changes in $fCO_2$ after accounting for the seasonal trend, are linked with salinity variations, with the highest regions of $fCO_2$ coinciding with the lowest salinity

regions and vice versa.  The inverse linear relationship between salinity and $fCO_2$ variations (Figure 8) suggests that mixing of estuarine and shelf water end members control the surface water distribution of $fCO_2$. Measured salinity agrees well with FVCOM model surface salinity fields at different stages of the tidal cycle. FVCOM output is used to estimate the temporal and spatial variations of $fCO_2$ in coastal waters. The model output demonstrates that neglecting the river plume leads to an overestimation of the magnitude of the $CO_2$ sink in the summer. The region may also be misidentified as a sink for

atmospheric $CO_2$ in the spring and autumn when it could be a weak source,  as suggested (Torres et al., 2020). Measurements from L4 are not representative of the estuarine zone and should not be scaled for this region.

The surface flux from estuaries and coastal waters has proven difficult to assess comprehensively (Borges et al., 2006). The high resolution $fCO_2$ data presented in this study show that tidal and estuarine-influenced coastal waters can quickly transition from a strong source to a strong sink depending on the state of the tide (Figure 11).  Inland and coastal waters are a

large uncertainty in global fluxes (Chen et al., 2013). The high resolution observations presented in this paper show that the head of the river/estuarine plume for this region is linked to the tidal state. This is critical for the accurate calculation of air–water flux from the region. Further studies are required to evaluate the wider applicability of this method to similar coastal regions. Distinctly different estuarine/coastal zones such as lagoons, river deltas and fjords will have their own distinct $fCO_2$ signatures and patterns driven in part by surface water circulation and mixing. High-resolution models that can resolve tides



and surface salinity in estuarine/coastal waters with high confidence, are a useful tool in ongoing and future efforts to constrain coastal air–sea $CO_2$ fluxes.

## 7. Acknowledgements

This research was supported by UK NERC via studentship (NE/L000075/1), the Shelf Sea Biogeochemistry pelagic research programme (NE/K002007/1), RAGNARoCC (NE/K002473/1), single centre national capability (CLASS; NE/R015953/1),

the Land Ocean Carbon Transfer project (LOCATE; NE/N018087/1), NERC grant NE/L007010 and the NERC LTSS National Capability Program that underpins numerical modelling work at PML. This research was also supported by EU projects MyCoast (EU Interreg Atlantic Area Program project MYCOAST (EAPA_285/2016)), IMMERSE (H2020 821926), AtlantOS (H2020 633211), FixO3 (FP7 312463) and RINGO (H2020 730944). We thank the captain and crew of the RV *Plymouth Quest* for all their assistance. We also thank Tim Smyth and Ming Yang for insightful discussions and data

relating to the WCO and PPAO time series and John Stevens and Rachel Beale for their help on the *PML Explorer*.

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

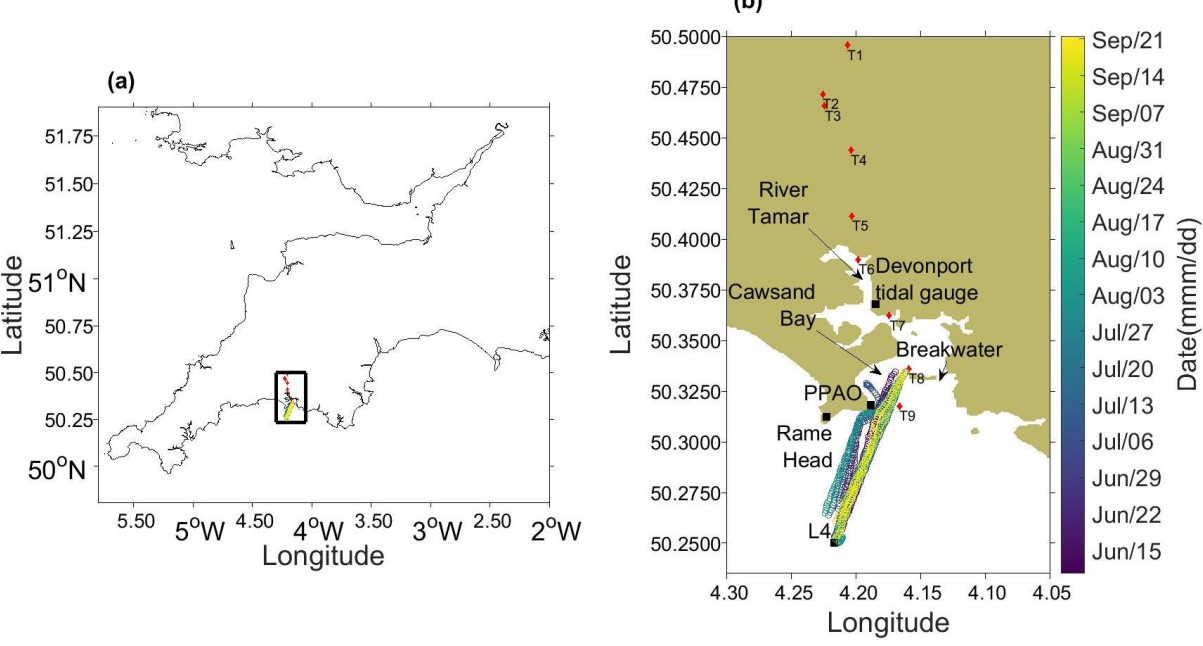


**Figure 1: (a) Map of the Southwest of the England Western Channel Observatory (WCO) and study area (black rectangle). (b) Breakwater to L4 transects for the 15 transects made with the RV *Plymouth Quest* (coloured circles). Ship tracks are coloured by date between 10th June   and 21st September   2016. The River Tamar, Devonport tidal gauge station, Plymouth Breakwater, Cawsand Bay, PPAO, Rame Head and Station L4 are marked. Transect sites T1–T9 are shown as red diamonds.**



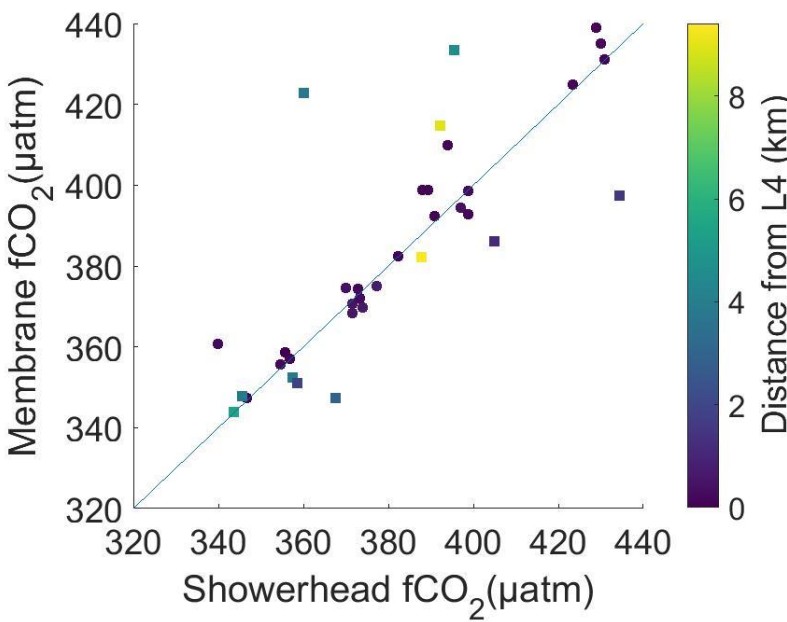


**Figure 2: Comparison of coincident seawater fCO₂ measurements made using membrane and showerhead equilibrator systems on the RV *Plymouth Quest*. Both systems sampled from the underway seawater supply during transects between L4 and the Plymouth Breakwater (filled squares). The ship was stationary for long periods (>1 hour) at L4 (filled circles). The membrane equilibrator system was switched at L4 to tubing connected to a near surface profiler (Sims et al., 2017) while the showerhead system continued**

**to sample from the underway supply. Data points are coloured as a function of distance from L4 (km). The 1:1 line represents perfect agreement and is shown for reference. Agreement between the two systems was best when close to L4 (RMSE = 1.49 µatm).**

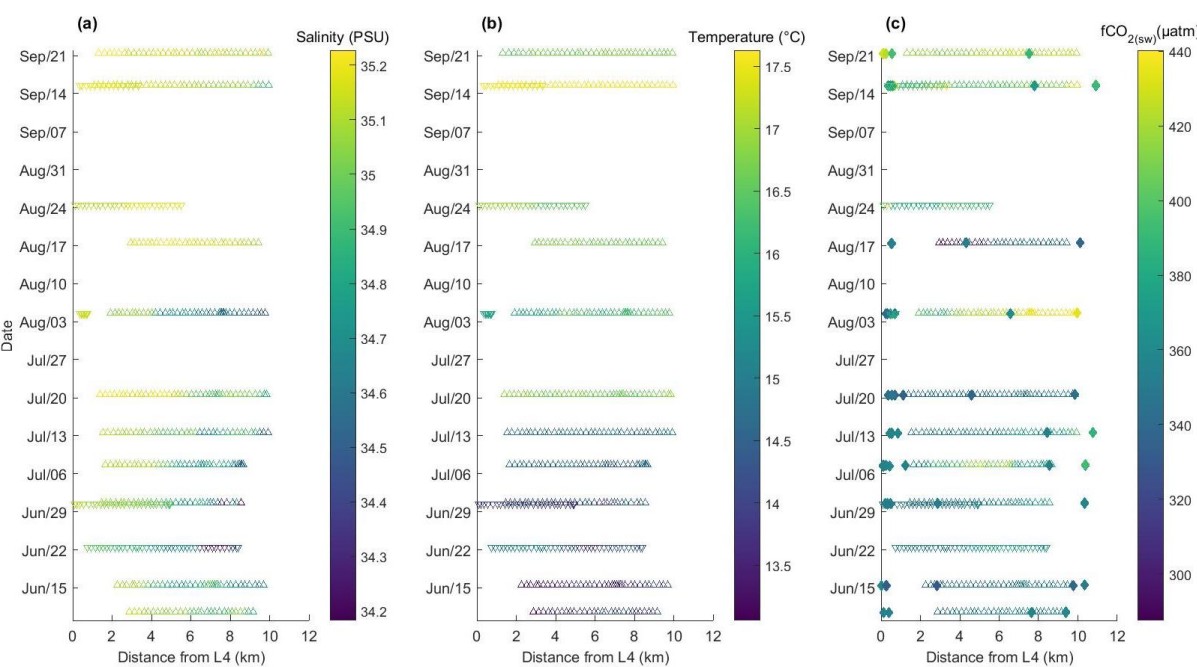





**Figure 3:** Surface water transects by RV *Plymouth Quest* between Plymouth Breakwater and station L4 from 10th June to 21st
September 2016. Plotted are (a) salinity, (b) temperature and (c) fCO$_2$ as a function of distance from L4. Outbound transects are
shown as downwards triangles and inbound transects are shown as upwards triangles for salinity, temperature and membrane
equilibrator fCO$_2$ (sampling period = 0.5 min). Showerhead equilibrator fCO$_2$ observations are denoted by filled diamonds.

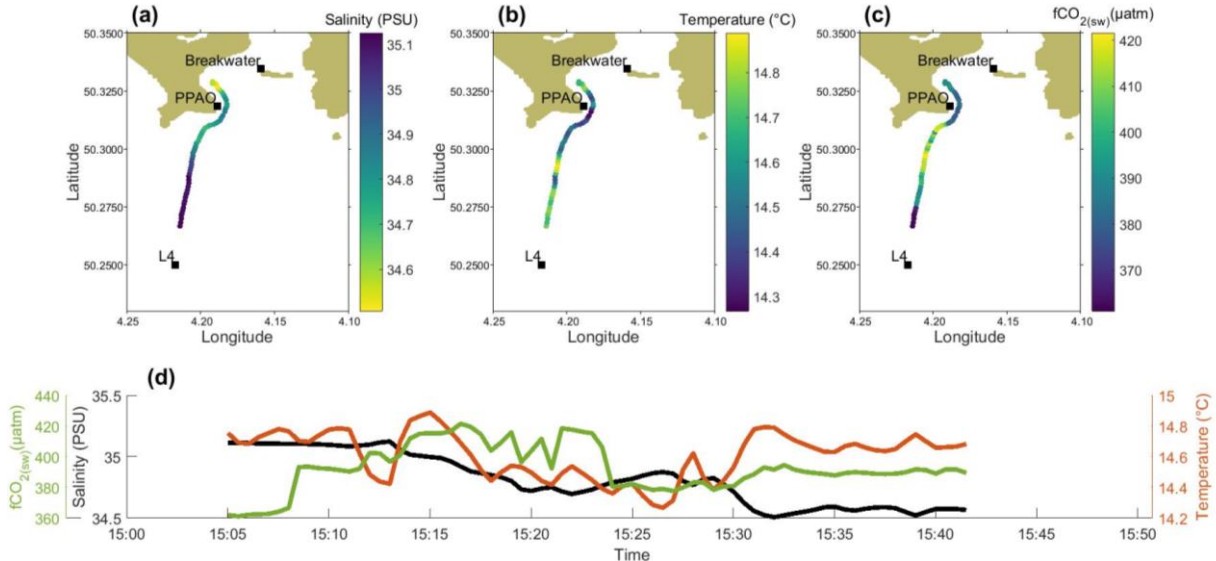

**Figure 4:** Example of a low water transect (transect mid-point = LW+0.5 hrs). Surface measurements of salinity, temperature and
fCO$_2$ represented spatially (a to c) and as a time series (d) from station L4 toward the coast. Data were collected onboard RV
Plymouth Quest on 7th July 2016 (filled circles, data every 30 sec).

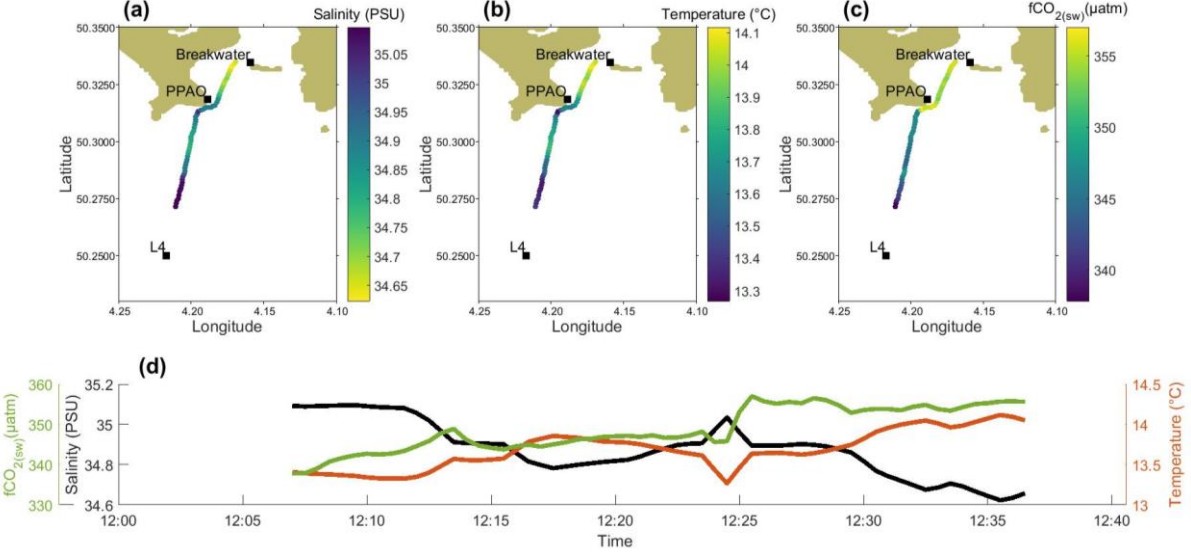

**Figure 5:** Example of a flooding tide transect (transect mid-point = LW+3.5 hrs). Surface measurements of salinity, temperature
and fCO$_2$ represented spatially (a to c) and as a time series (d) from station L4 toward the coast. Data were collected onboard RV
Plymouth Quest on 15th June 2016, (filled circles, data every 30 sec).






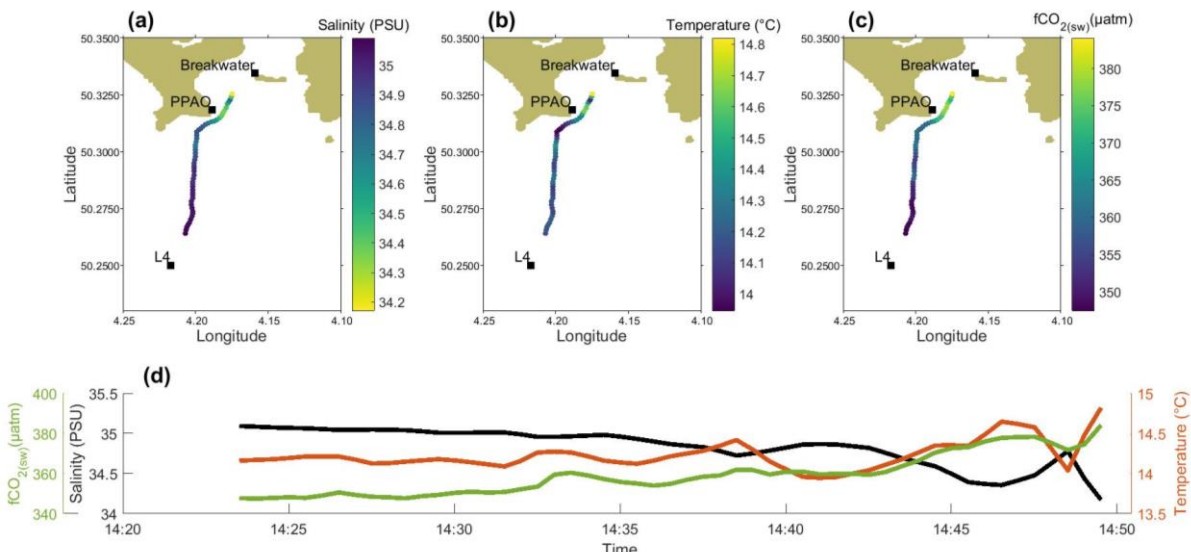

**Figure 6: Example of a high water transect (transect mid-point = LW+6 hrs). Surface measurements of salinity, temperature and fCO₂ represented spatially (a–c) and as a time series (d) from station L4 toward the coast. Data were collected onboard RV Plymouth Quest on 30th June 2016, (filled circles, data every 30 sec).**

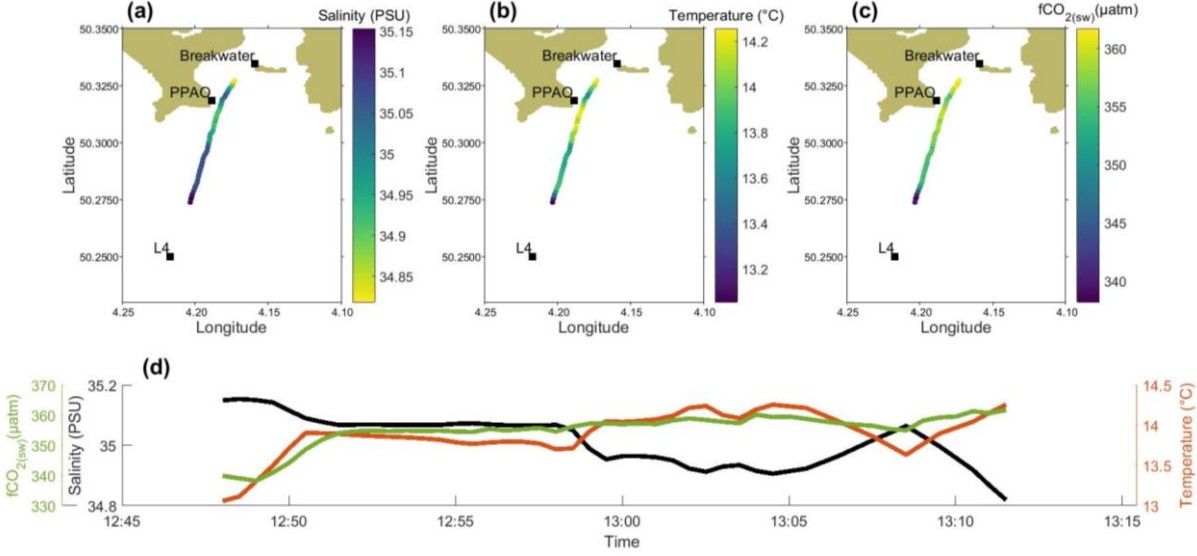


**Figure 7: Example of an ebbing tide transect (transect mid-point = LW+8.75 hrs). Surface measurements of salinity, temperature and fCO₂ represented spatially (a–c) and as a time series (d) from station L4 toward the coast. Data were collected onboard RV Plymouth Quest on 10th June 2016, (filled circles, data every 30 sec).**





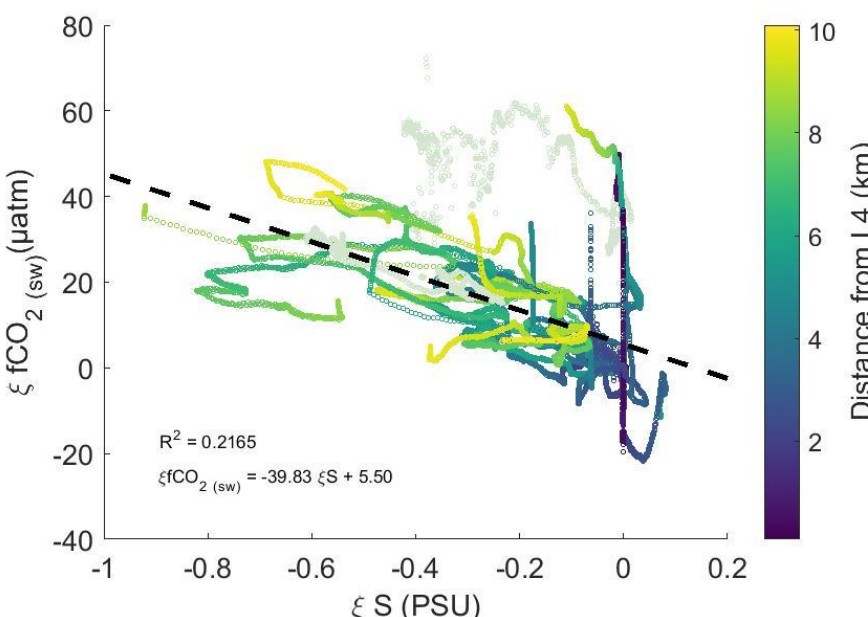

**Figure 8: Scatter plot of ξS versus ξfCO₂, difference of data from transects between Plymouth Breakwater and station L4 between 10th June and 21st September 2016. Data are coloured by their distance from station L4, the dashed black line is the best fit for the data. The data for 7th July is shown in light grey.**

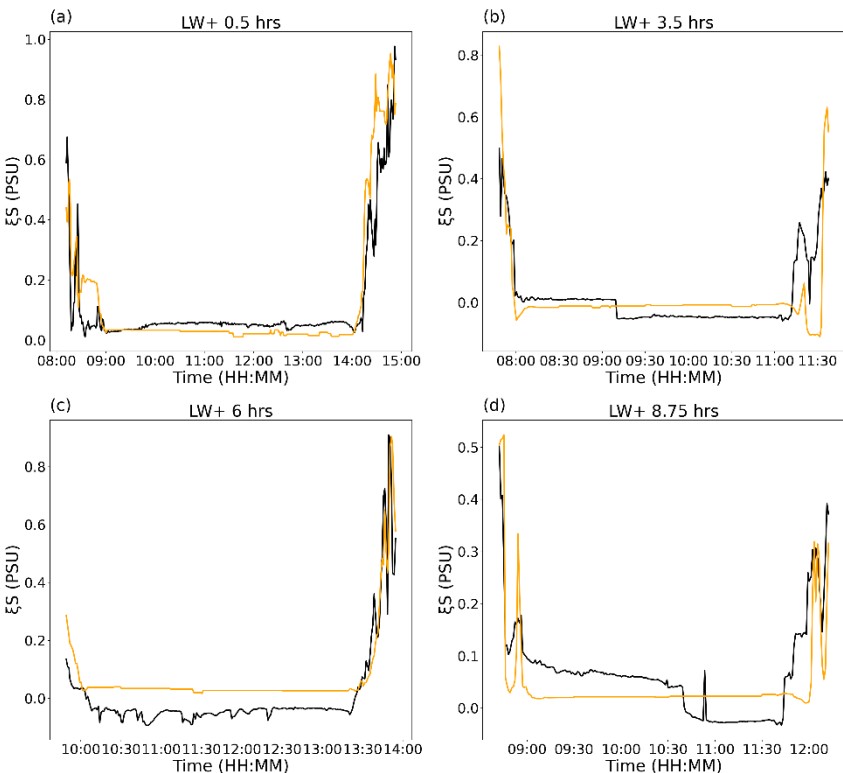





**Figure 9: Comparison of ξS from surface observations (underway RV Plymouth Quest, black) and the FVCOM model (orange).**
**Each of the four panels corresponds to the four example transects from four stages of the tidal cycle: (a) 7th July (LW+0hrs); (b) 15th June (LW+3.5hrs); (c) 30th June (LW+6hrs); and (d) 10th June (LW+8.75hrs).**

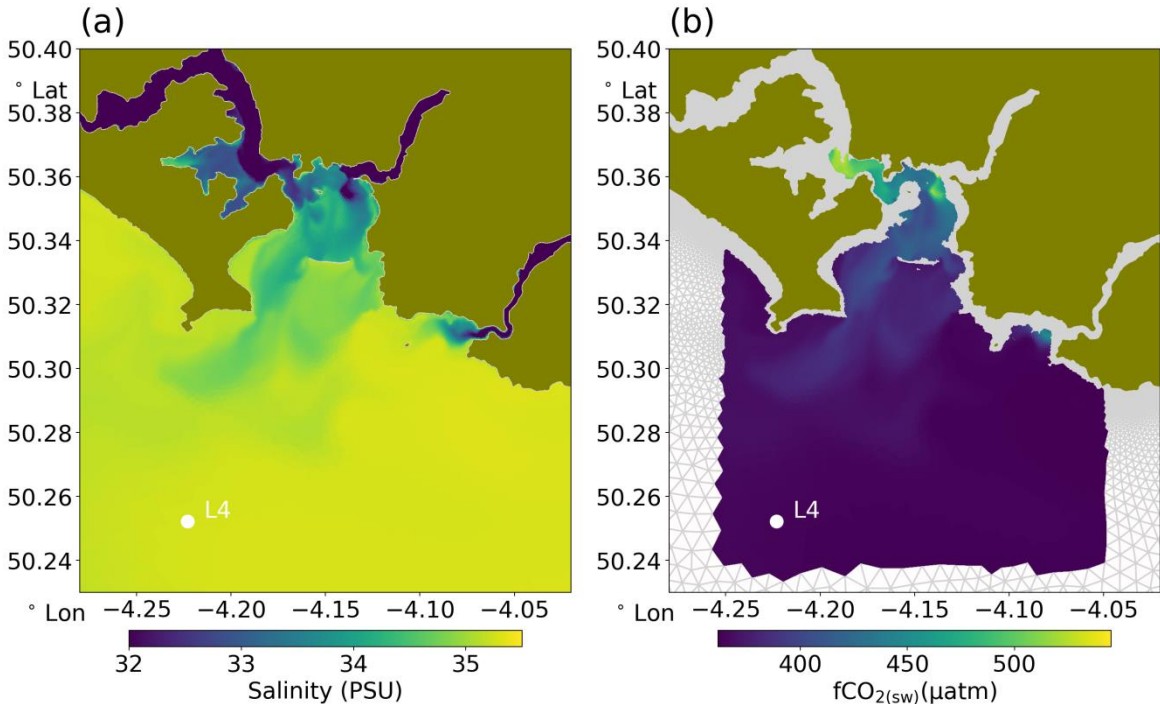

**Figure 10: (a) Model surface salinity field on 7th July at the midpoint of the observational transect. Note that the range of the colourbar was restricted to better depict salinity in the lower estuary and thus salinity in the upper estuaries is beyond the range of**
**the colourbar. (b) derived ξfCO₂ using determined ξS and ξfCO₂ relationship.**







**Figure 11: Modelled impact of a full tidal cycle (a) on tidal height (b) on air–sea CO$_2$ flux in the coastal zone (hourly calculations, negative fluxes are into the ocean) on 7$^{th}$ July 2016 with a fixed wind speed of 5.3ms$^{-1}$. The total integrated air–sea CO$_2$ flux within the modelled domain (c–f). Spatial variation in flux at four different stages of the 12 hour tidal cycle is shown: LW+0 hr (c). LW+3 hr (d), LW+6 hr (e). LW+9 hr (f). The red dots in (a) and (b) correspond to panels c–f.**


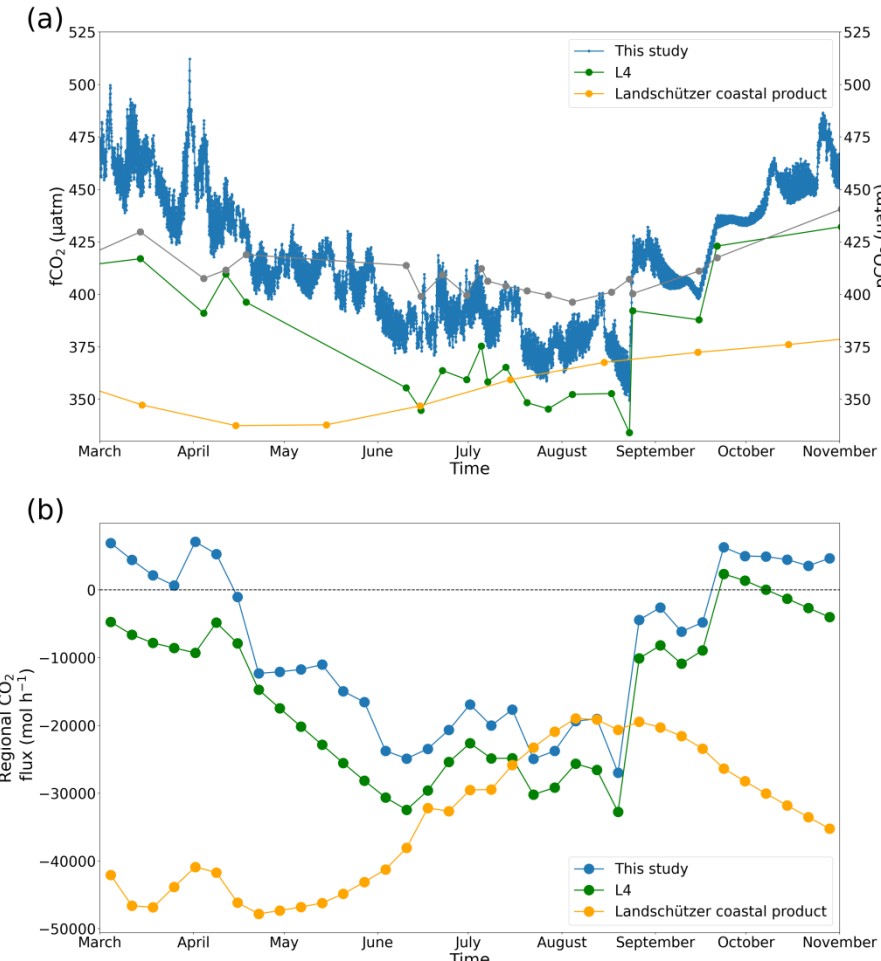

**Figure 12: (a) fCO$_2$ and pCO$_2$ values used to calculate regional CO$_2$ flux. Measurements of fCO$_2$ made by RV *Plymouth Quest* on station at L4 (L4, green dots). pCO$_2$ $_{(atm)}$ measured by Plymouth *Quest* on station at L4 (grey). The results of this study, hourly average fCO$_2$ across the valid model domain used for long term flux calculations (this study, blue dots). The monthly Landschützer pCO$_2$ data (Landschützer coastal product, orange dots) is from the closest valid node (50.125°N, 4.125°W) in the Landschützer et al. (2020) coastal data product. (b) Regional CO$_2$ flux calculated using the three different CO$_2$ over the valid model domain used for long term flux calculations (128 km$^2$, approximately same as in above figures). All fluxes were calculated using average monthly wind speed from the L4 buoy and using interpolated pCO$_2$ $_{(atm)}$ measured by RV *Plymouth Quest*. For this study fluxes were calculated across the domain using model derived fCO$_2$, temperature and salinity. The flux for the Landschützer and L4 measurements were calculated using temperature and salinity from the L4 buoy and scaled up to the equivalent area as this study. It is noted that the Landschützer node is outside of the model domain (which has a lower boundary of 50.24°N) and that the fluxes are made with pCO$_2$ rather than fCO$_2$.**