# Peer review of "Tidal mixing of estuarine and coastal waters in the Western English Channel is a control on spatial and temporal variability in seawater CO2"

_Biogeosciences, 2021_

## Author Response (AR1)

Dear Professor Middelburg

Thank you for your comments on our responses to the reviewer comments. We have re-examined the replies we submitted and on reflection can see that we did not clearly and explicitly state how we will address many of the reviewers' legitimate concerns. We appreciate that without more detailed replies and the edited manuscript it is not possible for you to determine whether we have addressed the reviewer comments appropriately. In addition to the revised manuscript that I will now upload, we elaborate on a number of our previous comments and provide additional detail on the implemented changes below.

We wish to highlight that the first reviewers concern regarding the RMSE of the model (which was stated as ~1 PSU in the original manuscript) should now be resolved. We have corrected our statistic for model performance so that it is only calculated for the part of the model domain we use to generate $fCO_2$, RMSE is dramatically reduced to 0.213 PSU. This makes a huge difference to the perceived accuracy of the model salinity and thus predicted $fCO_2$.

We now calculate a standard combined uncertainty in $fCO_2$ which is consistent with the International Bureau of Weights and Measures (BIPM) Guide to the expression of uncertainty in measurement (GUM) methodology. The standard combined uncertainty includes the uncertainty in the $\xi fCO_2$ and $\xi S$ relationship (18.95 µatm), uncertainty due to $\xi S$ (8.48 µatm) and the uncertainty in the $fCO_2$ observations (L4 fCO2 = 6.9 µatm, from the on station comparison, Figure 2). These uncertainties add in quadrature to give a standard combined uncertainty of 21.88 µatm. When this uncertainty is shown with the data in Figure 12 we are able to demonstrate that the differences in the average model $fCO_2$ and $fCO_2$ at L4 are not due to the uncertainty of our estimate.

We would like to highlight that the salinity-based correction is effectively a bias correction on the L4-based estimate of seawater $fCO_2$ concentration in the near coastal zone, and results in a worsening of the agreement between L4 and Landschützer estimates.. We inevitably have much more certainty in this correction during the period where there was more frequent sampling and have clearly stated this limitation in the revised manuscript.

The agreement between the two $pCO_2$ systems on station is due to the respective response times of the two systems. The $pCO_2$ agreement is better on station at L4 because the ship remains stationary and the water mass does not change drastically, which gives both systems enough time to fully achieve equilibrium. When the ship is moving across a heterogeneous environment, the membrane system (shorter response time) responds faster and can fully equilibrate; the showerhead system (longer response time) is unable to achieve full equilibration. The membrane equilibrator reflects the changes in $fCO_2$ in the different water masses, but the showerhead cannot do so in such a dynamic coastal environment. This is why the agreement between the two systems is much worse in transit then when at station L4.

The three $CO_2$ data sources we present in Figure 12a are all at different temporal scales, which presented some problems when calculating $CO_2$ flux. Calculating comparable $CO_2$ fluxes required a number of concessions, either averaging or interpolating, and the reviewers were right to point this out. Plotting the fluxes doesn't necessarily add much to the narrative of the paper, so we have decided to simplify Figure 12 by removing the bottom panel (12b) and all associated text.

Thank you for all your time and hard work on this submission.

Yours sincerely

Richard Sims

---

## Author Response (AR2)

**Report 1**

L42 : The fact that estuaries are heterotrophic and over-saturated in CO2 was shown for example by Frankignoulle et al. (1998) well before the cited reference.

We agree that this is a key reference. It is mentioned in the discussion and we have now added it to the introduction.

L47-54: I still don't understand what point the authors want to make here about "shallow-bottom boats" and why "restriction on large vessels" would be problem to accurately measure CO2 in estuaries. Is there a minimum threshold value of shaft depth or boat length to make accurate measurements of CO2 ?

The point we wish to make is that large ocean-going research vessels are unable to navigate shallow estuaries. Apologies that this was not clear. The following changes have been made to the paragraph:

Estuaries and rivers around major ports and harbours are deep and easily accessed with large research vessels. However, many estuaries are shallow and/or have irregular topography. Navigating shallow estuaries safely, particularly at low tide, is not always possible for many ocean-going research vessels, such that mapping the $p$CO$_2$ is challenging. The observations of $p$CO$_2$ included in SOCAT from upper estuarine waters come from a limited number of research vessels that are typically small (~25 m long) and have shallow (~3 m) drafts.

L70: please specify here that the transect is <8km long.

This change has been made.

L71: please replace English Channel by Western English Channel.

This change has been made.

L 87: The sentence "High river discharge rates in July 2006 were caused by several days of high rainfall and resulted in ~0.5 psu reduction in surface salinity at L4 (Rees et al., 2009)." seems out of place and is disconnected from the rest of the text.

We agree that this sentence is out of place. It has been edited and moved to the end of the paragraph:

L4 is seasonally stratified between late April and early October (Smyth et al., 2010b). The onset of stratification typically drives a diatom dominated spring bloom in early April. Nitrogen limitation later in the year favours a summertime dominance of smaller plankton (Widdicombe et al., 2010). A prominent feature of the coastal region around WCO is the coastal/tidal current that entrains buoyant freshwater from the River Tamar outflow with prominent frontal features (Uncles and Torres, 2013). The River Tamar is a large source of freshwater to the region despite being a relatively small river (Uncles et al., 2015). The coastal current moves along the west coast of Plymouth Sound adjacent to the Plymouth Breakwater and toward PPAO before following the coastline towards Rame Head peninsula (Uncles et al., 2015;Siddorn et al., 2003). The River Tamar is known to occasionally influence surface waters at the L4 station. For example, salinity reductions of ~0.5 have been observed after several days of rainfall led to elevated river discharge rates (Rees et al., 2009).

L393-400 this paragraph is disconnected from the rest of the discussion. I do not understand what's the take-home message. Do you mean that the Plymouth Quest transects are insufficient to exploit the PPAO EC flux data and that moorings with CO2 instrumentation are needed in the area to characterize the footprint area? I'm unsure how relevant this is for the discussion of a scientific paper, maybe leave this text for a proposal.

On reflection, we agree with the reviewer and have removed this paragraph from the manuscript.

**Report 2**

I also agree with the main conclusion that water mass mixing played a main role in controlling the estuarine pCO2 distribution and the air-sea CO2 flux in this system, though I do not like the complete discount of the role of biology. I felt and I feel a complete discount of biological control won't serve our understanding of this nice dataset well. At a minimum, the authors should explain why the very rapid and nearly vertical changes of pCO2 near L4 and away in some low salinity locations (Fig. 8) are not because of biological activities. I think this can be easily amended with one paragraph in the Discussion (I just don't understand why the authors chose to ignore this).

This is an oversight. The discussion made a very brief reference to plankton blooms, but we agree that it should be expanded upon. The paragraph now reads as follows:

The $fCO_2$ variability that is not explained by our relationship will be driven by other factors. Close to the coast such as in Cawsand Bay, these include gas exchange in shallow waters, excess river runoff during storms, $CO_2$ production in sediments, upwelling of water with different $CO_2$ concentration, and biological activity stimulated by nutrients in the Tamar plume. Biological processes in coastal ecosystems can strongly influence seawater $CO_2$ levels (Cai et al., 2020), which may explain some of the large $CO_2$ changes observed around L4 that are not linked to salinity changes. $fCO_2$ changes due to photosynthesis are typically a few µatm per day (Kitidis et al., 2019)) so large differences will likely have built up over time. Water masses with $fCO_2$ levels different to L4 may well be caused by factors such as biological activity, and then the water advected toward L4 by currents."

Cai, W.-J., Xu, Y.-Y., Feely, R. A., Wanninkhof, R., Jönsson, B., Alin, S. R., Barbero, L., Cross, J. N., Azetsu-Scott, K., and Fassbender, A. J.: Controls on surface water carbonate chemistry along North American ocean margins, Nature communications, 11, 1-13, 2020.

Lastly, I think the Introduction could be updated with a more recent status of coastal CO2 research (for example, it appears we are in the time of early 2000s; and not citing Burke Hales paper on a similar membrane equilibrator system). This won't diminish what you and your nice system have achieved.

Apologies for missing the Burke Hales reference, which is now included along with several other more recent studies of $CO_2$ research in the coastal zone:

Hales, B., Takahashi, T., and Bandstra, L.: Atmospheric CO2 uptake by a coastal upwelling system, Global Biogeochemical Cycles, 19, 2005.

Cai, W.-J., Feely, R. A., Testa, J. M., Li, M., Evans, W., Alin, S. R., Xu, Y.-Y., Pelletier, G., Ahmed, A., and Greeley, D. J.: Natural and anthropogenic drivers of acidification in large estuaries, Annual Review of Marine Science, 13, 23-55, 2021.

Ward, N. D., Bianchi, T. S., Medeiros, P. M., Seidel, M., Richey, J. E., Keil, R. G., and Sawakuchi, H. O.: Where carbon goes when water flows: carbon cycling across the aquatic continuum, Frontiers in Marine Science, 4, 7, 2017.

Shen, C., Testa, J. M., Li, M., Cai, W. J., Waldbusser, G. G., Ni, W., Kemp, W. M., Cornwell, J., Chen, B., and Brodeur, J.: Controls on carbonate system dynamics in a coastal plain estuary: A modeling study, Journal of Geophysical Research: Biogeosciences, 124, 61-78, 2019.

Roobaert, A., Laruelle, G. G., Landschützer, P., Gruber, N., Chou, L., and Regnier, P.: The spatiotemporal dynamics of the sources and sinks of CO2 in the global coastal ocean, Global Biogeochemical Cycles, 33, 1693-1714, 2019.

After 20 years of efforts in coastal waters, the third sentence is only correct in the sense of the high spatial and seasonal heterogeneity in coastal waters. (Estuarine and coastal water carbon dioxide (CO2) observations are relatively few compared to observations in the open ocean.) So modify it a bit or delete it.

We agree the sentence was not technically correct. The sentence and the one after have been deleted and the second sentence modified to read:

"The $CO_2$ flux from estuaries is uncertain due to the high spatial and seasonal heterogeneity of $CO_2$ in coastal waters."

I also suggest you modify at the end of line 21-22, adding something like "…fCO2 except a few locations(?at L4 as shown by Fig. 8 and within Cawsand Bay?) where influences of biological production are also clear". A complete discount of biological control won't serve our understanding of the dataset well.

We agree with the reviewer and have added the following sentence to the abstract to reflect the role that biology may be playing. "The correlation between salinity and $fCO_2$ was different in Cawsand Bay, which could be due to enhanced gas exchange or to enhanced biological activity in the region."

Please use italic font for p in pCO2 throughout the paper. The community uses italic p for "partial pressure" and p for -log unit (as in pH). As for f (fugacity), you are free to write either way as there is nothing to be confused of.

All instances of $pCO_2$ have now been changed to $pCO_2$.

L68, I am not qualified to question if there is any language issue here as the authors are all native English speakers (and I am not), but "Transects underway CO2 measurements" is odd to me. Should it be "Underway CO2 measurements along transects"?

This change has been made.

Methods:
For the membrane equilibrator, if I remember correctly, Burke Hales was the first one using it. Don't remember when he published it. Please search and cite him (if my memory is correct).

The original citation for the membrane equilibrator (Hales et al., 2004) is now included in the methods section.

Hales, B., Chipman, D., and Takahashi, T.: High-frequency measurement of partial pressure and total concentration of carbon dioxide in seawater using microporous hydrophobic membrane contractors, Limnol. Oceanogr.: Methods 2, 356-364, 2004.

Results:
L198: A comparison of two showerhead systems in another coastal system also showed about 3 ppm difference (Jiang et al. 2008).
Jiang, L.-Q., W.-J. Cai, R. Wanninkhof, Y. Wang, and H. Lu¨ger (2008), Air-sea CO2 fluxes on the U.S. South Atlantic Bight: Spatial and seasonal variability, J. Geophys. Res., 113, C07019, doi:10.1029/2007JC004366.

We have added this reference.

L265, change " in large rivers" to "in large river plumes" or "on large river influenced shelves".

This has been changed to "in large river plumes".

L278, It is not clear what does the author mean by "micro-environment". A semi-isolated small bay? Clarify a bit.

We agree that this was unclear. We have clarified by changing the terminology and adding a reference to the spatial scale in brackets. "As Cawsand Bay is a relatively small scale geographic feature (<1 km$^2$),"